# Photosynthetic, Molecular and Ultrastructural Characterization of Toxic Effects of Zinc in *Caulerpa racemosa* Indicate Promising Bioremediation Potentiality

**DOI:** 10.3390/plants11212868

**Published:** 2022-10-27

**Authors:** Simone Landi, Giorgia Santini, Ermenegilda Vitale, Gabriella Di Natale, Giulia Maisto, Carmen Arena, Sergio Esposito

**Affiliations:** 1Department of Biology, University of Naples “Federico II”, Via Cinthia, I-80126 Napoli, Italy; 2Department of Chemistry, University of Naples “Federico II”, Via Cinthia, I-80126 Napoli, Italy

**Keywords:** TEM, heavy metals, HSP70, marine algae, transporter, chloroplast

## Abstract

*Caulerpaceae* are unconventional green algae composed of multinucleated, single siphonous cells. The species of *Caulerpa* are acquiring major scientific interest for both their invasion in the Mediterranean ecological niche and for the production of valuable natural metabolites. Furthermore, the abilities of *Caulerpa* spp. in the biorecovery of polluted waters were recently investigated. Among heavy metal contaminants in marine systems, zinc (Zn) is considered a critical pollutant, progressively accumulating from plastic leachates. In this study, the responses of *Caulerpa racemosa* to different levels (5–10 mg L^−1^) of Zn were studied for 14 days under laboratory-controlled conditions. Effects of Zn were monitored by measuring the growth rate, photosynthetic efficiency and gene expression. Moreover, the ability of *Caulerpa* to remove Zn from seawater was monitored. Zn induced detrimental effects by decreasing the relative growth rate (RGR) and maximal PSII photochemical efficiency (F_v_/F_m_). Moreover, *C. racemosa*, grown in contaminated seawater, reduced the levels of Zn to a final concentration of 1.026 and 1.932 mg L^−1^ after 14 days, thus demonstrating efficient uptake. Therefore, our results characterized the effects of zinc on *C. racemosa* and the possible role of this alga as being effective in the bioremediation of marine seawater.

## 1. Introduction

*Caulerpa* spp. are a large and variegate group of green siphonous macroalgae widely distributed across tropical and temperate latitudes [1,2]. The presence of alien *Caulerpa* spp. has recently become an ecological emergence in the Mediterranean area, where the Australian-native *Caulerpa cylindracea* induced negative effects on the local marine fauna, landscape and ecology [3,4]. The most recognized consequence of the *Caulerpa* invasion is related to the fishery market: the increased presence of this alga modified the foraging habits of some Mediterranean-native fishes, particularly *Diplodus sargus* [4,5]. The occurrence in *Caulerpa* of high levels of the alkaloid caulerpin modifies the lipid metabolism of *D. sargus*, inducing detrimental effects on meat quality and nutritional value [6,7].

Nevertheless, *Caulerpa* spp. have been identified as able to produce a wide range of natural bio-products, exhibiting important medical and nutraceutical benefits [8]. In recent years, the production of bio-compounds useful for nutraceutical, pharmaceutical and cosmetic purposes by marine organisms has raised wide economic interest [9]. In particular, *Caulerpa* is able to synthesize interesting metabolites, such as caulerpin, sulfated polysaccharides, racemosin, glucans and lipids, which are highly desirable in biotechnology applications [3,8,10,11]. Promising immunostimulatory activities were obtained using different classes of polysaccharides from *Caulerpa.* These biomolecules increased the proliferation of macrophages, modulating their activities regarding phagocytosis and the production of nitric oxide and phosphatase, thus yielding effective results in in vitro tests against different types of diseases, such as carcinomas [11,12,13]. 

Furthermore, *Caulerpa* spp. recently offered promising results in seawater bioremediation [14,15]. This process is desirable in intensive aquaculture, a process producing a large amount of polluted wastewater. Marine seaweeds are particularly appreciated in seawater bioremediation and bioadsorption [16]. In particular, macroalgae are considered effective biofilters, because of their high growth rates, pliable reproductive modes and simple habitat requirements [14]. *Caulerpaceae* yielded interesting results in different bioremediation activities, namely basic dyes’ removal from waste streams, the treatment of seawater from aquaculture effluents and wastewater treatment to remove heavy metals [14,15,17,18,19,20,21]. This last process has become a critical challenge, due to the evidence that heavy metal pollution in aquatic systems poses a global pollution hazard [22,23]. 

Heavy metals are recalcitrant and represent persistent environmental pollutants [20,22,23]. In this context, macroalgal bioremediation has acquired increased value [14,15]: bioremediation methods are widely used due their effectiveness, low cost and environmental protection. Specifically, biosorption is the process of sequestering heavy metals by non-living biomass of algae, lichens, fungi, mosses, bacteria and biomass wastes, including rice husks, wood fibers, sawdust, fly ash and tea stems [20,24,25,26].

Among heavy metals, zinc (Zn) represents an essential element for macroalgae, microalgae and higher plants [22,27,28]. Zn deficiency induces serious consequences in plants and seaweeds, including a reduction in photosynthetic yields and nitrogen uptake, but its excessive accumulation alters different physiological and metabolic processes [20,22,29,30]. Autotrophs require an optimal Zn supply for correct growth and development, but in the last few years, the increasing amount of Zn-rich waste in terrestrial and aquatic environments has induced the hyperaccumulation of this metal [23,27,29]. An excess of Zn in the aquatic environment affects different cellular processes in aquatic macro- and microorganisms [31,32,33]: reduced growth rates have been reported in a number of algae, diatoms and cyanobacteria, resulting in oxidative stress and severe inhibition in yields [32,33]. In particular, Zn excess induced a reduction in cyanobacteria biomass, protein and carbohydrate content [33]. Furthermore, in strains of *Microcystis, Anabaena, Spirulina* and *Synechococcus*, Zn contamination caused detrimental effects on chlorophyll a content and the photosynthetic process [31,33,34]. In addition, diatoms and microalgae are different marine microorganisms significantly influenced by Zn [35]. Among these, species such as *Thalassiosira pseudonana*, *Skeletonema marinoi*, *Skeletonema costatum*, *Isochrysis galbana* and *Dunaliella tertiolecta* showed a significant growth reduction when cultivated in seawater supplemented with Zn [36,37]. Zn accumulation in aquatic environments is mainly caused by its intensive use in plastics and rubber production [38]. Different studies revealed, as major sources of Zn contaminants in seawater, leachates from new and aged plastics and car tire rubber [31]. In addition to these wastes, anthropogenic pollutants such as antifouling agents and atmospheric aerosols have been reported as possible additional sources of this element in the marine environment [38]. 

The aim of this work was to investigate, for the first time, the adaptation of a specific *Caulerpa* species (*C. racemosa*) to adverse conditions represented by an excess of Zn and its potentiality in the biorecovery of contaminated water. The different physiological, structural and biochemical parameters in *Caulerpa* subjected to Zn were analyzed and Zn levels in seawater monitored after *Caulerpa* growth. Expression analysis was conducted on key enzymes to identify the regulation of *Caulerpa* metabolism.

## 2. Results

### 2.1. Zinc’s Effects on Caulerpa racemosa’s Growth and Photosynthesis

Growth rates and photosynthesis were analyzed to investigate the ability of *C. racemosa* to grow under adverse conditions such as Zn-contaminated seawater. *C. racemosa* was exposed to different Zn treatments: low-Zinc (ZnL, 5 mg L^−1^) and high-Zinc (ZnH, 10 mg L^−1^). Globally, both levels induced detrimental effects on *C. racemosa* growth (Figure 1). Interestingly, ZnH induced the rapid short-term growth of *C. racemosa* compared with both the control (seawater, Sw) and ZnL. After 7 days, ZnL showed a significant reduction in *C. racemosa* fresh weight (a 1.2-fold decrease compared with Sw), while no significant differences were observed in ZnH and Sw. Long-term treatments with Zn (up to 10 days) induced a fresh weight reduction in *C. racemose* in both ZnL (-1.77-fold) and ZnH (-2.01-fold) compared with Sw. Specific differences in the relative growth rate (RGR) were observed (Table 1). *Caulerpa* samples cultivated in Sw showed RGR% equal to 3.26 and 5.08, respectively, after 7 and 14 days. *Caulerpa* ZnL showed a significant reduction in the RGR%, changing from 1.98 to 1.01 after 7 and 14 days. Consistently with the fresh weight, *Caulerpa* ZnH showed a higher RGR% after 7 days compared with *Caulerpa* ZnL of around 2.97, while we observed a lower RGR% of around 0.81 after 14 days (Table 1).

As shown in Figure 2, different photosynthetic parameters, namely PSII maximal photochemical efficiency, Fv/Fm and quantum yield of PSII electron transport (fPSII), were significantly influenced by both Zn treatments. After 7 days, a reduction in F_v_/F_m_ value in ZnL (−1.68) and ZnH (−2.25) and in fPSII in ZnL (−1.73) and ZnH (−2.83) were recorded compared to SW, respectively. After 14 days, no further significant changes in F_v_/F_m_ and fPSII values were evidenced compared to the previous 7 days in Zn-treated algae. More specifically, comparing ZnL and ZnH with Sw, the reduction in the F_v_/F_m_ ratio was −1.94 and −2.75, while the reductions in φPSII were −1.97 and −3.44, respectively. Contextually, the highest non-photochemical quenching (NPQ) value in these thalli at both 7 and 14 d compared to Sw and ZnL was found. In particular, ZnL showed an increase in NPQ after 14 days of Zn treatment of around +2.17, while ZnH increased significantly the NPQ after 7 days (+2) and after 14 days (+3.59). 

### 2.2. Ultrastructural Properties of Caulerpa racemosa

Chloroplasts of *Caulerpa* in Sw showed quasi-spherical shapes, the dense aggregation of thylakoids concentrically appressed to a large central starch grain and several lipid formations, possibly plastoglobuli, typically surrounding the central starch grain (Figure 3a–c). In algae exposed to ZnL conditions, chloroplasts appeared similar to those observed in Sw conditions, except for the lack of plastoglobuli (Figure 3d), although some of them showed some disorganization in the grana structure (Figure 3e,f). Starch granules appeared normal (Figure 3d,e). In algae exposed to ZnH, the major disaggregation of thylakoids was observed (Figure 3g–i). Furthermore, the chloroplasts appeared longer—instead of spherical—and thylakoids appeared more distant from each other, with scarce grana organization. Starch granules were less organized and plastoglobuli were scarcely present. Furthermore, lipid-containing peroxisomes and disorganized mitochondria could be noted (Figure 3i,k).

### 2.3. Zn Uptake by C. racemosa from Seawater

We investigated the ability of Zn uptake in *C. racemosa* by analyzing the Zn levels in seawater before and after *Caulerpa* growth. As shown in Table 2*, C. racemosa* is able to efficiently accumulate Zn, thus reducing metal levels in seawater. At the end of the experimental period, Zn levels were 1.03 mg L^−1^ in ZnL and 1.93 mg L^−1^ in ZnH, showing a severe reduction in Zn levels of −79.5% and −80.7%, respectively. 

### 2.4. Expression Analysis of Key Genes of Caulerpa Metabolism

Considering the significant effects of high Zn concentrations on *Caulerpa* growth and photosynthesis, we investigated the molecular expression of genes involved in different aspects of algal metabolism. In particular, we analyzed genes coding for enzymes involved in the biosynthesis of proteins involved in the protection from heavy metals, namely phytochelatin (g8088) and heat shock protein 70 kDa (HSP70—g1266), and genes coding for Zn transporters, g7326 and g224. Furthermore, we investigated the expression of the cytosolic isoform of glucose-6-phosphate dehydrogenase (cytG6PDH—g2472), which is involved in different aspects of glucose metabolism and the response to abiotic stress. As shown in Figure 4, all analyzed genes showed an increase in expression under the ZnH condition compared with Sw. In particular, *Caulerpa* treated with Zn showed a +4.3-, +2.9-, +13.2- and +2.4-fold increase when compared with the control (Sw) for phytochelatin, Hsp70 and Zn transporters g7326 and g224, respectively. On the other hand, cytG6PDH was expressed in both Sw and ZnH but showed no significant differences between the treatments (Figure 4E). 

## 3. Discussion

Renewed interest has been recently reported in the utilization of algae in the detoxyfication of polluted waters. Heavy metal adsorption is a convenient process to achieve environmental protection, especially considering the rapid regeneration and growth of algae. Different groups of marine macro- and microorganisms were investigated as potential resources for bioremediation, namely cyanobacteria, diatoms, microalgae (e.g., *Chlorella* sp., *Chlamydomonas* sp.) and macroalgae (e.g., *Laminaria japonica*, *Ulva lactuca*) [9,39,40,41]. These organisms show the efficient uptake of heavy metals from polluted waters and have thus received considerable research attention due their wide range of applications [42]. Under laboratory conditions, the rate of heavy metal sequestration by most macro- and micro-algae can reach more than 80%, and, in some cases, 95% [39,40]. In this context, *Caulerpa* sp. would be an interesting organism for heavy metal removal from seawater. Different works reported the ability of dried and processed *Caulerpa* in heavy metal uptake [20,21,43]. Pavasant et al. [20] used the dried biomass of *Caulerpa lentillifera* for the biosorption of a wide range of heavy metals, including Cd^2+^ and Pb^2+^, highlighting the effective use of this macroalga in the removal of heavy metals from low-strength wastewater. Metal uptake abilities were also reported for *Caulerpa serrulata:* samples of this alga were functionalized using ethylenediamine, displaying the capability to remove lead (Pb), copper (Cu) and cadmium (Cd) from polluted waters [21]. Biomasses of *Caulerpa fastigiata* were used for Pb removal in toxic aqueous solutions as well, reporting the best biosorption capacity of up to 16.11 mg g^−1^ of Pb^2+^ on *Caulerpa* biomass [43]. 

In this work, we report the ability of vital *C. racemosa* to remove excess Zn from seawater. Although an excess of Zn improves the growth rate in the first few days of cultivation, the experimental conditions induced medium- and long-term detrimental effects, resulting in a decrease in both growth and photosynthesis. Usually, *Caulerpa* species are significantly affected by cultural parameters, namely light, temperature, salinity and nutrient availability [1,44,45,46]. These parameters modify growth rates, photosynthetic parameters, chlorophyll content, nitrogen uptake and antixodiant activity [44,45,46,47]. 

Our results reported that in *C. racemosa*, high levels of Zn (10 mg L^−1^) induced a booster effect in the first few days, making it possible to achieve an RGR of 2.97% after 7 days, a value close to that of the controls (3.26%). Despite this initial benefit, prolonged growth in the presence of high Zn induced a reduction in RGR and the photochemistry of PSII (F_v_/F_m_ and ΦPSII). On the other hand, the thermal dissipation processes remained low at each growth condition. This result is not surprising, since the growth irradiance is very low (namely 40 PPFD) and does not induce the activation of an effective non-photochemical quenching mechanism. Consistently, detrimental effects on photosynthetic metabolism were reported when cultivating *Caulerpa lentillifera* in the presence of an excess of arsenite. This species showed the downregulation of the expression of those genes encoding the most important photosynthetic components [48]. 

Furthermore, the ultrastructural appearance of *C. racemosa* may reflect the intracellular stress induced by Zn exposition. Usually, heavy metals induce damage to membranes, a reduction in cell wall surface and loss of the selective permeability of cell compartments [26,49]. Our results reported an alteration in the ultrastructure of the chloroplasts of *Caulerpa* grown in the presence of Zn, suggesting that the presence of the metal modifies the development of chloroplasts in a concentration-dependent manner. This behavior has been previously observed for other bioindicators, such as moss and liverwort [50,51]. The damages on chloroplasts could be induced by the oxidative burst induced by Zn accumulation, and the consequent overproduction of ROS. These processes would induce the lipid peroxidation of cell membranes and severe injury to thylakoids. When exposed to ZnL conditions, chloroplasts showed a strong decrease in plastoglobuli, suggesting that Zn may affect the lipid metabolism in these organelles. *Caulerpa* cells subjected to the ZnH condition showed worse damage to chloroplasts and cellular membranes. In particular, the thylakoidal system lacks grana, possibly indicating a strong decrease in PSII complexes, which can be usually found in the grana regions. *Caulerpa*
*sertularioides* showed ultrastructural effects on the absorption surface of this species subjected to different Cu treatments [52]. Interestingly, ultrastructural analysis was also performed on biomasses of *Caulerpa* used for metal biosorption [43]. These analyses revealed the presence of heterogeneous surfaces and an increased number of pores with different diameters, likely improving the uptake of metal ions [43].

However, *C. racemosa* is able to survive under the negative effects of an excess of Zn, showing an adquate photosynthetic rate and maintaining the uptake ability of this element for up to 14 days. Different authors reported the significant detrimental effects of Zn in a wide range of algae, microalgae and diatom species, including a reduction in cell division and viability, growth inhibition and a reduction in chlorophyll content [32,33,35]. Comparison with other macroalgae, namely *Cladophora, Sargassum thunbergii* or *Ulva fasciata*, indicated similar or reliable reductions in RGR values compared to *Caulerpa* after comparable prolonged Zn treatments [22,53,54]. In addition, after the experimental period, a severe reduction in Zn was observed in the seawaters used to grow *Caulerpa*. Consistenstly, similar abilities were reported for *C. lentillifera* in terms of nitrogen uptake [45]. In fact, the biomass and specific growth rate of *C. lentillifera* increased—and then decreased—according to the fluctuation in the NH_4_/NO_3_ ratio, suggesting this species as being suitable for the bioremediation of acquaculture wastewater [45].

The reduction in Zn concentration in seawater reported in this manuscript was probably regulated by *Caulerpa*’s metabolic activities. Our research reported an upregulation of genes coding for enzymes involved in protection against heavy metals, namely phytochelatin and HSP70, and Zn transporters. Consistently, a previous spectroscopic analysis on the same species revealed the ability to remove hydrocarbons from seawater guided by metabolic functions [55]. In particular, HSP70 is one of the most important protein families involved in the response to abiotic stresses, acting as a molecular chaperon and ensuring the correct protein folding [56,57]. This family plays a critical role in the response to heavy metals in different photosynthetic organisms, from algae to moss and higher plants [48,49,50,58]. The activation of HSP70 together with phytochelatin highlighted the ability of *Caulerpa* to counteract the toxic effects induced by Zn. This activation was recently reported both in *Caulerpa* and in other algal species (e.g., *Dunaliella salina*) subjected to different heavy metal stress, namely copper or arsenic, making this behavior a specific marker of stress also for micro- and macroalgae [47,58]. To monitor the redox metabolism, we also analyzed the expression of cyt-G6PDH, which showed a critical role in NADPH supply [57,59]. Cyt-G6PDH showed no significant increase in expression under the ZnH condition, highlighting minor effects in terms of oxidative stress induced by the excess of this metal for *Caulerpa*. Nonetheless, the expression of cyt-G6PDH demonstrates the vitality of *Caulerpa* also after prolonged stress induced by Zn treatment. Finally, the increase in the expression of two different Zn transporters (g7326 and g224) substantially demonstrated the effective and active role of *Caulerpa* in the removal of Zn from seawater. In particular, g7326 showed a reliable increase in expression; this gene is indicated in the *Caulerpa* genome database (https://marinegenomics.oist.jp) as orthologous to the *Arabidopsis* transporter At3g08650, as zinc nutrient essential 1 (AtZNE1). Interestingly, *Arabidopsis* knock-out plants for AtZNE1 showed growth defects under excess Zn or Fe deficiency, but no effects on the total Zn and Fe content were noted [60]. Therefore, a crucial adaptation role under a manipulated concentration of Zn or Fe was identified for this transporter [60]. Based on the major contribution of g7326 in the response to Zn, it has been possible to hypothesize a similar role also in *Caulerpa*. 

## 4. Materials and Methods

### 4.1. Caulerpa Cultivation 

Thalli of *Caulerpa racemosa* J. Agardh 1873 were collected from natural populations in the Napoli gulf in September 2021. Samples were transported immediately to the laboratory using boxes filled with seawater. For laboratory experiments, healthy thalli were selected and rinsed several times with sterilized seawater. The environmental average zinc concentrations in seawater are 0.6–5 ppb [61]. Samples of *C. racemosa* were divided into three different groups and grown in natural seawater (Sw) and in natural seawater with the addition of ZnSO_4_ 5 mg L^−1^ (ZnL) and of ZnSO_4_ 10 mg L^−1^ (ZnH). These Zn levels were selected on the basis of published experiments performed on different seaweed species [22,62,63]. Three biological replicates were used for each treatment. Algae were grown using an irradiance of 40 μmol photons m^−2^ s^−1^ and a photoperiod of 16:8 h in a light/dark cycle. Portions of *C. racemosa* were sampled after 7 and 14 days and immediately frozen in liquid nitrogen. The growth and survival of *Caulerpa* samples were monitored daily. Fresh weight was measured at the beginning of the experiment (W0), regularly during the entire experimental period and at the end (Wt). The relative growth rate (RGR %) was calculated using the equation given in Choi et al. [64]: RGR% day^−1^ = 100 ln (Wt/Wo)/t, where “t” represents the number of days.

### 4.2. Photosynthesis 

Chlorophyll a fluorescence measurements were made using a pulse amplitude-modulated fluorometer (Junior-PAM, Walz, Germany). On 30-min dark-adapted leaves, the background fluorescence signal, F_0_, was induced by internal light provided by a blue LED of around 2–3 μmol photons m^− 2^ s^−1^, at a frequency of 0.5 kHz. An optic fiber of 1 mm diameter was inclined at 45° at a distance of 0.5 mm from samples kept in water suspension. The maximal fluorescence level in the dark-adapted state (F_m_) was measured under a 1s saturating light pulse (3000 μmol photons m^−2^ s^−1^) at a frequency of 10 kHz; the maximal PSII photochemical efficiency (F_v_/F_m_) was calculated as F_v_/F_m_ = (F_m_ − F_0_)/F_m_. Under illumination at 40 photosynthetic photon flux densities (PPFD), the steady-state fluorescence (Fs) was measured, and the maximum fluorescence (Fm’) in the light-adapted state was determined by applying a saturating pulse of 0.8 s with over 2000 μmol photons m^−2^ s^−1^. The quantum yield of PSII electron transport (fPSII) was calculated as (Fm′-Fs)/Fm’ according to Genty et al. [65], while the non-photochemical quenching (NPQ) was expressed as (Fm-Fm′)/Fm′, as reported in Bilger and Björkman [66].

### 4.3. Zinc Quantification

The seawater samples were previously and appropriately diluted before analyses. The concentrations of Zn were measured through inductively coupled plasma mass spectrometry (ICP-MS Aurora M90, Bruker). After optimizing the ICP-MS, Zn concentrations in the samples were determined by comparing results with a calibration curve. The range of linearity of the calibration curve, the limit of detection, the limit of quantification and the coefficient of variation for Zn were 1 ppb–1000 ppb; 0.3 ppb; 1 ppb; 2.35%, respectively. In order to ascertain the accuracy of the measurement, an analysis of reference materials was carried out using 1 multi-element standard (ICP-MS Multielement Mix 30 elements—Ultra Scientific IMS-120) and 2 internal standards (Indium Standard for ICP—Ultra Scientific ICP-149; Yttrium Standard for ICP—Ultra Scientific ICP-139). All the reference materials used were certified according to ISO 17034.

### 4.4. Microscopy

Algal fixation was carried out with a 3% (*v*/*v*) glutaraldehyde solution in a phosphate buffer (pH 7.2–7.4) for 2 h at room temperature. Post-fixation was performed with buffered 1% (*w*/*v*) OsO4 for 1.5 h at room temperature. Dehydration employed ethanol up to propylene oxide and was followed by embedding into Spurr’s epoxy medium. A Philips EM 208 S TEM was employed for observation, as previously described [26].

### 4.5. RNA Extraction and qRT-PCR

The RNA was extracted from 100 mg of fresh *Caulerpa* from Sw and ZnH samples using TRizol reagent (Life Technologies, Carlsbad, CA, USA). RNA extraction was performed on three biological replicates for each treatment. The RNA amount was measured using a NanoDrop ND-1000 spectrophotometer (Thermofisher, Waltham, MA, USA). cDNA syntheses were performed using the ThermoScript RT-PCR System. The gene expression analysis was carried out via qRT-PCR. Platinum SYBR Green qPCR SuperMix (Life Technologies, Carlsbad, CA, USA) was used. Gene expression in ZnH was calculated using Sw samples as calibrator controls; GAPDH served as an endogenous reference gene [48]. The quantization of gene expression was carried out using the 2^−ΔΔCt^ method, as in Livak and Schmittgen [67]. The mRNA amount was calculated in each sample, relative to the calibrator sample, for each corresponding gene. 

### 4.6. Statistics

Each experiment was performed in at least three replicates. Values were expressed as mean ± standard deviation (SD). The statistical significance of growth parameters, photosynthetic parameters and zinc content between the different Zn concentrations and and controls (sea water) was calculated through analysis of variance (ANOVA, calculations correspond to α = 0.05). Differences between means were evaluated for significance using the Tukey–Kramer test. 

## 5. Conclusions

This manuscript reported the effective ability of *C. racemosa* to remove Zn from seawater. Different concentrations of Zn were assayed, and, in both cases, we observed a significant reduction in Zn in seawater used for *Caulerpa* cultivation. Although the selected concentration of ZnH stimulated a booster effect in the first few days of cultivation, both ZnL and ZnH induced long-term negative effects on *C. racemosa* growth rates and photosynthesis. TEM microphotographs revealed ultrastructural damages on chloroplasts, identifying these development effects on these organelles as suitable markers for abiotic stress in *C. racemosa*. Expression analysis indicated the effective regulation of *Caulerpa* metabolism to respond to the Zn stress and the activation of transporters able to improve the uptake of this element. Furthermore, the adverse effects induced by Zn were ameliorated by the activation of protection proteins, namely phytochelatin and HSP70. Among the analyzed Zn transporters, g7326 appears to be the most important enzyme involved in Zn uptake in *Caulerpa*. 

Despite these findings, the ability of C. *racemosa* to survive to these conditions suggests that this macroalga promisingly responds to Zn pollution and could be an interesting candidate for biomonitoring and biorecovery activities in seawater.

## Figures and Tables

**Figure 1 plants-11-02868-f001:**
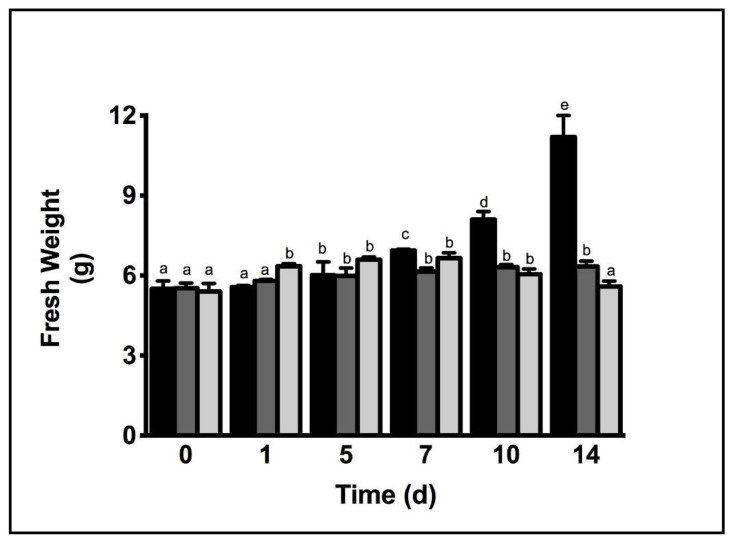
Fresh weight of *Caulerpa* grown in seawater (Sw—black bars), seawater + ZnSO_4_ 5 mg L^−1^ (ZnL—dark grey bars) and seawater + ZnSO_4_ 10 mg L^−1^ (ZnH—light grey bars) for 14 days. Letters indicate significant differences between treatments and days (*p* ≤ 0.05). Letters indicate ANOVA significance between treatments.

**Figure 2 plants-11-02868-f002:**
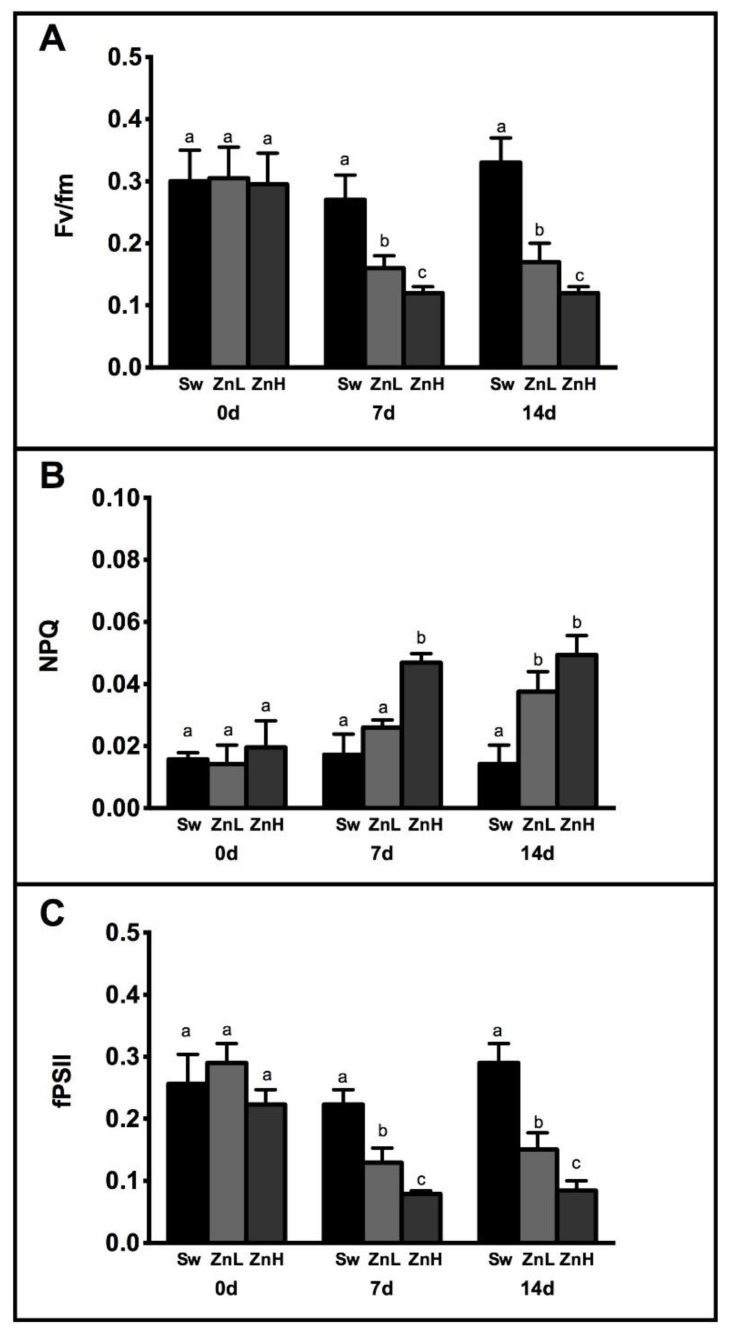
Fv/Fm (**A**), NPQ (**B**) and fPSII (**C**) of *Caulerpa* grown in seawater (Sw—black bars), seawater + ZnSO_4_ 5 mg L^−1^ (ZnL—light grey bars) and seawater + ZnSO_4_ 10 mg L^−1^ (ZnH—dark grey bars) for 14 days. Letters indicate ANOVA significance between treatments.

**Figure 3 plants-11-02868-f003:**
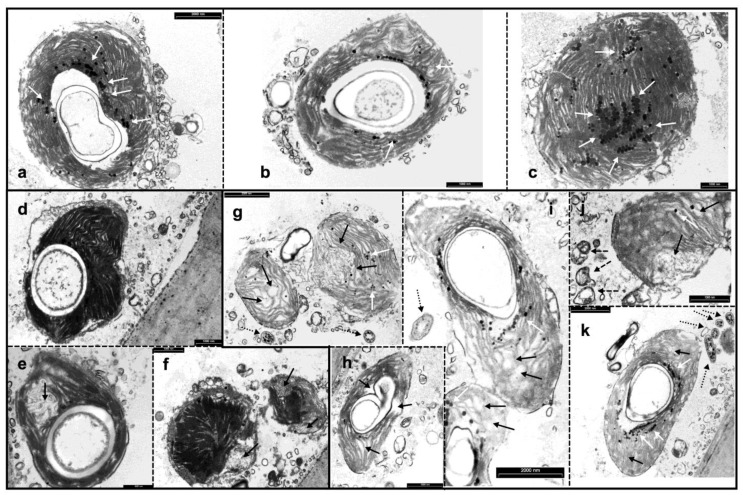
TEM micrographs of chloroplasts from *C. racemosa* samples exposed to Sw (seawater) (**a**–**c**); ZnL (ZnSO_4_ 5 mg L^−1^) (**d**–**f**); and ZnH (ZnSO_4_ 10 mg L^−1^) (**g**–**k**). Description: (**a**) and (**b**) typical *Caulerpa* chloroplasts, with a central starch granule and well-organized thylakoids; white arrows indicate plastoglobuli; (**c**) *Caulerpa* chloroplast with many plastoglobules (white arrows); in this section, the starch granule is not present (or not visible); (**d**) *Caulerpa* chloroplast exposed to ZnL conditions; the organelle appears healthy but no plastoglobulus is visible; (**e**) *Caulerpa* chloroplast exposed to ZnL conditions, in which the severe disorganization of thylakoids is evident (black arrow); (**f**) two chloroplasts from ZnL-exposed *Caulerpa*; base membrane disorganization is indicated by black arrows; (**g**) ZnH-exposed Caulerpa chloroplasts; the strong disorganization of the thylakoids is indicated by black arrows, and rare plastoglobuli are visible (white arrows); (**i**) this ZnL chloroplast shows a central starch granule, but the thylakoid system appears disorganized (black arrows), even more plastoglobuli are visible (white arrows) and a peroxisome with lipid drops inside is indicated (dotted arrow); (**j**) another ZnH chloroplast with unstacked membranes (black arrows) and rare plastoglobuli (white arrows); some unhealthy mitochondria are nearby visible (dashed arrows); (**k**) a ZnH-exposed chloroplast similar to that in (**i**); note several microbodies (possibly peroxisomes) with lipid drops inside (dotted arrows). Scale bars: (**a**,**j**,**k**) 2000 nm; (**b**–**h**)1000 nm; (**e**) 500 nm.

**Figure 4 plants-11-02868-f004:**
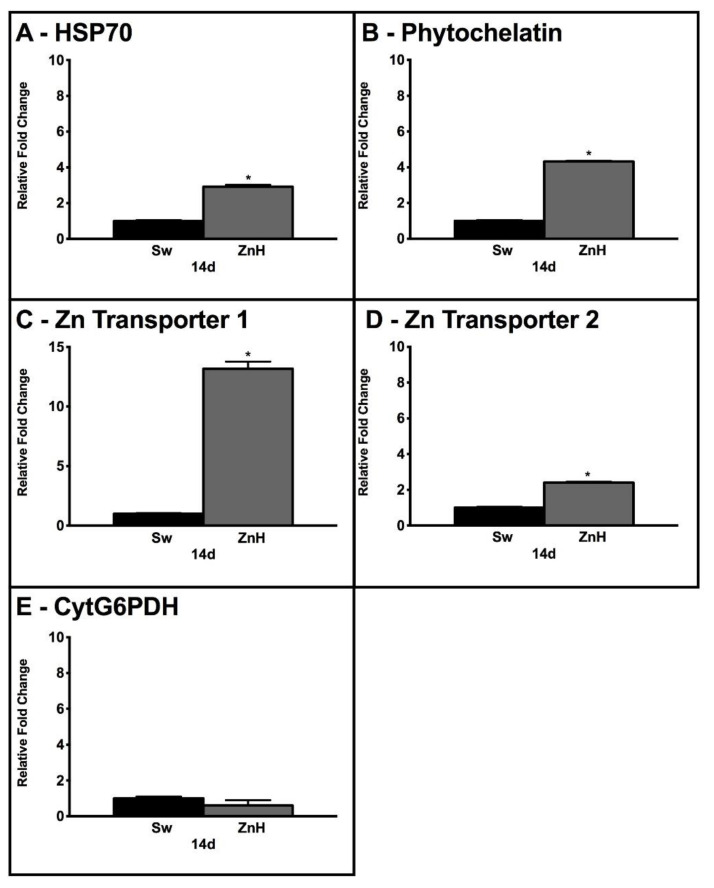
Gene expression levels of HSP70 (**A**), phytochelatin (**B**), Zn transporter 1 (**C**), Zn transporter 2 (**D**) and cytG6PDH (**E**) after 14 days in Sw (black bars) and ZnH (grey bars) conditions, measured by qRT-PCR. Variations are indicated as relative fold change in ZnH with respect to respective controls (Sw). mRNA levels were calculated relative to the expression of the RNA GAPDH, used as a calibrator. Asterisks indicate significantly different values at *p* ≤ 0.05 (*).

**Table 1 plants-11-02868-t001:** Relative growth rate (RGR%) of *Caulerpa* grown in seawater and in presence of ZnSO_4_ at different concentrations. Uppercase (7d) and lowercase (14d) letters indicate significant differences between treatments at corresponding days (*p* ≤ 0.05).

	Sea Water	ZnSO_4_ 5 mg L^−1^	ZnSO_4_ 10 mg L^−1^
RGR% 7d	3.26 ± 0.09 a	1.98 ± 0.05 b	2.97 ± 0.08 c
RGR% 14d	5.08 ± 0.12 A	1.00 ± 0.01 B	0.81 ± 0.01 C

**Table 2 plants-11-02868-t002:** Zinc content in seawater used to cultivate *C. racemosa*. Asterisks (*) indicate significant differences between treatments comparing T14 vs. T0 (*p* ≤ 0.05).

	Sea Water	ZnL T0	ZnH T0	ZnL T14	ZnH T14
**ZnSO_4_ (mg** **L^−1^)**	<0.001	5.0 ± 0.001	10.0 ± 0.001	1.03 ± 0.03 *	1.93 ± 0.07 *

## Data Availability

Not applicable.

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
