# Peer review of "Photosynthetic, Molecular and Ultrastructural Characterization of Toxic Effects of Zinc in Caulerpa racemosa Indicate Promising Bioremediation Potentiality"

_plants, 2022, doi:10.3390/plants11212868_

Round 1

Reviewer 1 Report (Previous Reviewer 1)

Dear Editor,

Authors have addressed most of the quarries, so, the article now acceptable for publication.

Kind Regards

Iftikhar

Author Response

Thank you for the peer-review process. 

Reviewer 2 Report (Previous Reviewer 2)

In this first revision, the authors made important changes in the manuscript that show an improvement of the content. However, it remains unclear what the objectives of this study are, bioremediation and/or toxicity? The authors extended the response studies of this algae, but the Title indicates only bioremediation. The Title must be indicative of the content (bioremediation and toxicity studies). Other comments:

1) Lines 72-80: Taking advantage of the addition of this new paragraph, it would have been desirable for the added references to be more specific to the marine environment (only one is related). In addition, the authors used 5 and 10 mg/L of Zn. Are these concentrations environmental? Giving data and references on Zn levels in marine environments would be interesting.

2) Line 94: ZnL is repeated, ZnH?

3) Table 1: The comparison should be within the same day of exposure. For example, lowercase letters for day 7 and uppercase for day 14. In this case, it is not very appropriate to compare two exposure days far apart.

4) Lines 142 and 147: It is not necessary to include the name of the species descriptor here.

5) The methodology related to gene expression is not clearly explained and developed in Material and methods. This has to be corrected for the manuscript to be published.

6) Unit expressions do not have a uniform format, or exponential format, or fractional format. Check.

7) No matter how many references the authors give, the fresh weight is not an adequate measure due to its enormous variability.

8) The format of the references is not uniform, and there are species names without italics. Check the bibliography properly.

Round 2

Reviewer 2 Report (Previous Reviewer 2)

The changes suggested in the manuscript have been considered correctly. From my point of view, the manuscript can be published.

This manuscript is a resubmission of an earlier submission. The following is a list of the peer review reports and author responses from that submission.

Round 1

Reviewer 1 Report

Dear Editor,

Authors can find detail comments or suggestion in an attached pdf file. 

Kind regards

Reviewer 2 Report

This manuscript studies the toxic effect of zinc on the alga Caulerpa racemosa. A simple removal study of this metal is also considered. In general, the scientific investigation in this manuscript is very simple, it is a typical toxicity test. There is no significant contribution to the knowledge of toxic effects. There is also no in-depth study of the characteristics of this material as a biosorbent. In my opinion, the manuscript does not have enough content to be published in this journal.

Some considerations that authors should take into account are:

1) Title: What are the objectives of this work? Toxicity or bioremediation? The Title does not refer to bioremediation (only toxicity). It is necessary to take into account that this manuscript is intended to be published in a special issue on xenobiotic elimination. Only in the text of the manuscript does it seem that the authors give a little more importance to bioremediation.

2) Figure 1: If the comparisons are between treatments, the letters would have to go from a, b and c, always three by three for each day (there cannot be letters e and d since there are only three treatments). It seems that the authors compared the effect of each day for the same treatment but not between treatments.

3) Figure 1: Y-axis legend, the abbreviation for gram is g or gm.

4) Lines 182-192: this paragraph is not related to the content of the manuscript.

5) Line 217: How do the authors know that the alga was alive? Also, in a biosorption process, metal removal can be by adsorption and bioaccumulation. How do the authors demonstrate the degree of bioaccumulation?

With the methodology used, it is not possible to draw conclusions in this regard.

6) If the authors consider that this algae is resistant to the effects of zinc (comparison with other algae could be a good reference), it would be interesting to study tolerance mechanisms. Are Phytochelatins Involved?

7) Line 239: The fresh weight is not a very adequate measure of growth due to the great variation of these measures due to the different retention of water that the living material may have, this is further aggravated with algae. How has this fact been controlled? Justify the use of this procedure.

8) The biomass quantification procedure is not very clear. How was this procedure done? Were the algae weighed each day and returned? how was the water removed?

9) This formula does not seem correct; for the result of the growth rate to be a %, it is necessary to know a reference growth rate value. The growth rate only has units of 1/time. Don't multiply by 100.

10) How were the bioremediation experiments carried out?

11) The authors use ANOVA, but have the postulates of the ANOVA been verified? There would be three groups with only three results per group (the triplicate). Is this enough to be able to apply a parametric test?

12) Lines 277 and 279: Why do the authors use two different post hoc tests?

13) Conclusions: the conclusions should be more concrete. Do not include results in the conclusions.

14) In the bibliography there are species names that are written without italics. In addition, the format of the bibliography is not uniform.